# Measuring and Disentangling Ambiguity and Confidence in the Lab

**Daniela Di Cagno [1] and Daniela Grieco [2,*]** 

[1]  Department of Economics and Finance, Luiss Guido Carli, 00198 Rome, Italy; ddicagno@luiss.it
[2]  Department of Economics, Bocconi University, 20136 Milano, Italy
*   Correspondence: daniela.grieco@unibocconi.it; Tel.: +39-338-462-8960

**Abstract:** In this paper we present a novel experimental procedure aimed at better understanding the interaction between confidence and ambiguity attitudes in individual decision making. Different ambiguity settings not only can be determined by the lack of information in possible scenarios completely "external" to the decision-maker, but can also be a consequence of the decision maker's ignorance about her own characteristics or performance and, thus, deals with confidence. We design a multistage experiment where subjects face different sources of ambiguity and where we are able to control for self-assessed levels of competence. By means of a Principal Component Analysis, we obtain a set of measures of "internal" and "external" ambiguity aversion. Our regressions show that the two measures are significantly correlated at the subject level, that the subjects' "internal" ambiguity aversion increases in performance in the high-competence task and that "external" ambiguity aversion moderately increases in earnings. Self-selection does not play any role.

**Keywords:** ambiguity; confidence; competence; precision; self-selection

## 1. Introduction

Several real-life decisions have to be taken on the basis of probability judgments where the information needed by the decision-maker is partially or totally missing. This lack of information can derive from the ambiguity of the possible scenarios, odds, and payoffs, or it can result from ignorance on the individual's absolute performance or relative position as compared to other individuals' characteristics or performance. In the former situation, the type of ambiguity affecting the decision-making process is generated by instances that are external to the individual. In the latter case, ambiguity derives from the difficulty the individual experiences when evaluating her own capabilities or traits or knowledge and is therefore strongly connected with her degree of (absolute or relative) confidence. When it is the performance that matters, the decision to take part in a task can change dramatically according to self-evaluation: people are rarely well-calibrated and often show "over-" or "under-confidence", and the presence of both biases is likely to modify the perceived degree of ambiguity affecting their decisions. Moreover, when individuals have to self-evaluate with respect to peers, this potential confidence bias is exacerbated by the additional difficulty of estimating peers' performance or characteristics or knowledge and of individuating one's own reference-group.

This paper experimentally addresses the issue of ambiguity and confidence jointly within a unique experimental framework. In particular, it aims at disentangling the effects of internally-generated and externally-generated ambiguity situations on individual decision making by controlling for self-assessed levels of competence.

As for confidence, the experimental design allows the authors to measure the three types of overconfidence introduced by Moore and Healy (2008)—overestimation, overplacement and overprecision [1]. As for ambiguity, the subjects' ambiguity attitude is measured by using both a 'willingness-to-bet' paradigm (choice between lotteries) and a 'willingness-to-invest' paradigm (investment choice) in order to control for framing effects. For the first time to the best of our knowledge, a Principal Component Analysis is performed to examine the behavioral consequences of internally-generated and externally-generated ambiguity. A regression analysis shows that the measure of internal and external ambiguity aversion are significantly correlated. However, while the latter positively depends on performance, the former positively depends on earnings and negatively depends on the perceived ease of the experiment.

Very few studies address the issues of ambiguity and confidence jointly. Both phenomena are not easy to measure, might present unclear responses to monetary incentives, interact with risk-attitude, and could be strongly context-dependent. An exception is represented by Brenner et al. (2011)'s paper on financial decisions in which the authors derive a model based on the max-min ambiguity framework [2] that links overconfidence to ambiguity aversion and predicts that overconfidence is decreasing in ambiguity (i.e., the more a portfolio's returns are ambiguous, the less overconfident investors are). Their experimental findings support this prediction. In a lab experiment with MBA students, Shyti (2013) [3] investigated the choices of overconfident decision makers when dealing with uncertain options whose outcomes have different likelihoods, finding that only overconfident decision makers choose less uncertain options for low-likelihood outcomes, thus, not matching the standard Prospect Theory predictions. The relationship between ambiguity and confidence has also been studied in the context of gender differences. In a financial decision context, Gysler et al., (2002) [4] found a significant and positive relationship between individual overconfidence and competence measures that produces opposite effects according to the gender of the subject. In a framework aimed at reproducing trading activities, Yang and Zhu (2016) [5] show that traders who think they are on average better in terms of trading ability trade more when prior information about the distribution is ambiguous, with a stronger effect for males.

Recent studies on confidence in one's own knowledge (Blavatskyy, 2009) [6] have emphasized the need to provide the subjects with proper financial incentives in order to elicit their actual beliefs on one's capabilities: despite the popularity of elicitation of confidence intervals, some authors (e.g., Cesarini et al., 2006) [7] have shown that this methodology might cause the deliberate misreporting of confidence intervals due to strategic reasons. Furthermore, risk aversion (and in general risk attitudes) might "dramatically affect the incentives to correctly report the true subjective probability of a binary event, even under Subjective Expected Utility" ([8], p. 1).

A possible way to relate ambiguity and confidence is interpreting confidence as an "internal" source of ambiguity (affecting an individual's decisions), as opposed to "external" sources where the lack of information is not about the decision-maker and her own characteristics. This is coherent with Abdellaoui et al. (2011) [9] who define a source of uncertainty as concerning a group of events that is generated by a common mechanism of uncertainty[1].

The way we relate ambiguity to confidence is also in line with Attanasi, et al. (2014)'s belief elicitation under ambiguity [11], where ambiguity is generated in a two-stage setting à la Klibanoff, et al. (2005) [12] by relying on different "external" sources in the first stage (i.e., binomial distribution, uniform distribution, unknown distribution) at a between-subject level. Each external source is generated by a common mechanism of uncertainty at a within-subject level. Then, given the second-stage urn composition (which is unknown), the subjects' beliefs about it are elicited. The heterogeneous distribution of beliefs they obtain

---

[1]　In Ellsberg's (1961) [10] classical two-color paradox, the source of uncertainty can be represented by the color of a ball drawn randomly from an urn containing 50 black and 50 red balls (the known urn), or by the color of a ball drawn randomly from an urn with 100 black and red balls in unknown proportion (the unknown urn). Alternatively, other sources of uncertainty could be Stock indexes like the Dow Jones index or the Nikkei index (with the foreign index implying a higher level of uncertainty for a US resident due to the "home bias").

can be interpreted as an "internal" source of ambiguity, i.e., different levels of "confidence" for different subjects. They find no correlation between this "confidence" and the subjects' behavior under ambiguity.

Overconfidence is a well-established bias where subjective confidence in one's judgment is systematically stronger than objective accuracy. Despite the fact that most prior research has treated different types of overconfidence as one and the same, this distortion takes different forms according to the way accuracy in judgment is measured and categorized, as subjects are rarely overconfident in every confidence category. A comprehensive classification has been introduced by Moore and Healy (2008) [1], who suggest three categories: *overestimation*, which occurs when people think they are better than they actually are; *overplacement*, which we observe when people think they are better than their peers; *overprecision*, which happens when people think they are better in judging their performance than they actually are or than their peers.

To the best of our knowledge, although there exist some experimental attempts to infer individual confidence about one's performance (as it will be reported below in detail) and to measure overestimation, overplacement and overprecision, none of them has been implemented in order to measure their effects on risk or ambiguity attitudes. On the contrary, two articles that have studied the effects of overconfidence on *risk* attitudes. Both these studies focus on voting choice, modeling it as a risky lottery. Attanasi at al. (2014) study subjects' overconfidence and its relation with risk aversion in a theory-driven experiment where each subject can get an informative signal about the others' voting preferences [13]. Attanasi et al. (2017) [14] explicitly refer to Moore and Healy (2008) [1], and show how to incorporate overprecision into an expected utility framework and derive predictions about the preferred majority threshold of a risk-averse and overconfident (over-precise) decision maker.

## 1.1. Overestimation

Overestimation typically emerges when subjects are badly-calibrated in assessing their own absolute performance in a task or in a set of tasks. Literature in psychology provides evidence in favor of systematic miscalibration and hard-easy effect (e.g., Lichtenstein and Fischhoff, 1977; Juslin et al., 2000; Merkle, 2009) [15–17]. A typical non-incentivized assessment of beliefs asks subjects to predict their performance or their probability of success. A common finding is that on questions perceived as easy (where the success rate is high), the average confidence is substantially lower than the actual success rate, whereas in questions perceived as hard (with a lower success rate), the average confidence is substantially higher than the actual success rate: presenting subjects with easy vs. hard tasks is a typical way to induce under- vs. overconfidence in the lab.

Recent incentivized measurements in economic experiments have revealed different patterns. Blavatskyy (2009) [6] does not directly elicit estimation measures, but infers underconfidence in the sense of underestimation from the choice of a payment scheme: either one question is selected at random and the subject receives a payoff if she answers this question correctly, or the subject receives the same payoff with a stated probability set by the experimenter to be equal to the percentage of correctly answered questions (although the subject does not know this is how the probability is set). The majority choose the second payment scheme, which she interprets as reflecting underestimation.

Urbig et al. (2009) [18] elicited confidence about one's performance over a set of ten multiple choice quiz questions using an incentivized mechanism that relies on probability equivalents for bets based on one's performance: their findings show almost no miscalibration since the average elicited probability equivalent is extremely close to the actual rate of success.

Both Blavatskyy (2009) [6] and Urbig et al. (2009) [18] noted the difference between their findings and those from the earlier psychology literature, and speculate that the difference may be due to the introduction of incentivized elicitation devices.

Clark and Friesen (2009) [19] studied the subjects' confidence via a set of tasks using either non-incentivized self-reports or quadratic scoring rule (QSR) incentives, including two types of real effort tasks involving verbal and numerical skills. They found underestimation more prevalent than overestimation and better calibration with incentives, with underestimation being stronger among

those where higher effort is needed. One potential limitation of their analysis, however, is that unless the subjects are risk neutral, QSR may result in biased measurements of confidence (we return to this point below in more detail).

Murad et al. (2016) [20] used two elicitation procedures: self-reported (non-incentivized) confidence and an incentivized procedure that elicits the certainty equivalent of a bet based on performance: the former reproduces the "hard-easy effect", whereas the latter produces general underconfidence, which is significantly reduced, but not eliminated when the effects of risk attitudes are filtered out.

## 1.2. Overplacement

Overplacement deals with relative position and social comparison. In his seminal work, Festinger (1954) [21] posited that people experience difficulty in testing their own ability against an objective standard and therefore reduce this uncertainty by using the abilities of others as the subjective reality. Overplacement has been observed in several domains where the "unrealistically high appraisals of one's own qualities versus those of others" emerges (Belsky and Gilovich, 1999 [22]); research on overplacement has not only been categorized as research on 'overconfidence'[2] but also on the 'better-than-average' effect. Subjects who overplace themselves are asked to rate their characteristics or performance in relative terms with respect to peers, and typically rate themselves as better than average. The vast majority of studies measures overconfidence without providing subjects incentives to assess their evaluation correctly (for a review of this literature, see Alicke and Govorun, 2005 [24]).

Research presenting the most impressive findings of overplacement has tended to focus on simple domains, such as driving a car or getting along with others (College Board, 1976–1977 [25]; Svenson, 1981 [26]). Among the studies that measure overconfidence on how individuals' beliefs on their performance translate into the probability of winning, Camerer and Lovallo (1999) [27] infer overplacement from subjects' decision to enter a market of defined capacity when survivors have to be relatively better than their peers in a task they self-selected into. Grieco and Hogarth (2009) [28] investigate participants' choice of betting on their own relative performance in a task or on a 50/50 risky lottery without knowing how well they did.

## 1.3. Overprecision

Overprecision can be defined as "the excessive faith that you know the truth" (Moore et al., 2019 [29]) and leads us to rely on our own judgment too much, despite its many flaws (ibidem). Among the practical economic consequences, the empirical evidence supports the failure to protect from risks (Silver, 2012 [30]), the neglect of others' perspectives and the failure to take advice (Minson and Mueller, 2012 [31]), too high a willingness to trade (Daniel et al., 2001 [32]), and too little a search for ideas, people and information (Haran et al., 2013 [33]). This is a robust[3] and well-documented phenomenon—although the less studied form of overconfidence—but lacks a full explanation and is measured in such a way that can lead to biases. In a typical psychological study of calibration, a participant answers a number of questions or makes a series of forecasts about future events and, for each item, expresses a subjective probability that the chosen answer or forecast is correct. A person is considered well-calibrated if there is a precise match between subjective assessments of likelihood and the corresponding empirical relative frequencies.

Laboratory studies of overprecision typically use two paradigms for eliciting beliefs. The former is the "Two Alternative Forced Choice Approach" (Griffin and Brenner, 2004 [34]): subjects choose between two possible answers to a question and indicate the probability that the chosen answer is correct. Empirical evidence (e.g., Kern, 1997 [35]) shows that subjective confidence is imperfectly

---

[2]  Although Kwan et al. (2004) emphasized the need of distinguishing between the two manifestations [23].
[3]  There are few, if any, documented reversals of overprecison, whereas there are many documented reversals of the other two varieties of overconfidence (overestimation and overplacement).

correlated with accuracy, i.e., when people are confident, their confidence is not justified by accuracy. The latter is the "Confidence-Interval Paradigm" (Alpert and Raiffa, 1982 [36]): subjects have to specify confidence intervals, namely state whether they were 98% (or between the 25th and 75th fractiles) sure that an event occurred or is occurring[4].

One of the challenges associated with the measure of overprecision in judgment is a shortage of an incentive-compatible scoring rule (Moore et al., 2013 [38]). A measure alternative to the ones described above is based on incentive compatible choices instead of beliefs. The classic elicitation method for assessing precision in judgment—the 90% confidence interval—is not incentive-compatible since respondents will make their intervals infinitely wide if you reward them for high hit-rates (i.e., getting the right answer inside their intervals) and infinitely narrow if you reward them for providing narrower intervals; rewarding both makes the calibration of the two rewards difficult. Jose and Winkler (2009) [39] proposed an incentive-compatible scoring rule for continuous judgment: they ask the subject to specify a specific fractile (let us say the 10th fractile of their subjective probability distribution of Barack Obama's body weight), i.e., to estimate a fractile in a subjective probability distribution. The weakness of this approach lies in the complexity of the payoff formula that is difficult to understand for most subjects.

Our experimental design allows us to measure these three "types" of overconfidence in an incentive-compatible way conceived for each specific definition of overconfidence and tailored to permit a joint investigation of overconfidence and ambiguity.

Ambiguity (or 'Knightian uncertainty') refers to situations where information is insufficient to pin down easily probabilistic beliefs about the external events that might affect the outcome of a decision. Ambiguity and uncertainty are often used as synonyms (Etner et al., 2012 [40]), although some scholars distinguish between the two concepts according to the extent of available information (uncertainty being closer to ignorance), or use the term ambiguity when information is unavailable from a subjective (but not an objective) point of view. In any case, they are both opposed to risk that represents "probabilized" uncertainty.

The earlier literature on how people react to ambiguity was surveyed in Camerer and Weber (1992) and in Camerer (1995) [41,42]. Some more recent articles on models of ambiguity-sensitive preferences and empirical (mostly experimental) tests were reviewed by Etner et al. (2012) [40].

For what concerns this paper, we are interested in the way ambiguity has been reproduced in the lab and how ambiguity attitudes have been measured. The experimental literature advocates the use of incentivized elicitation tasks and suggests a diversity of designs to measure ambiguity preferences: choices (pair-wise or multiple choices lists, used to measure the "willingness to bet") between a risky and an ambiguous lottery, choices (pair-wise or multiple choices list, an alternative way to measure the willingness to bet) between an ambiguous lottery and a sure amount of money, monetary valuation (willingness to pay) of risk and ambiguity lottery, and allocation/investment questions.

Abdellaoui et al. (2011) [9] captured attitudes towards ambiguity by involving a choice-based probabilities approach, using the traditional 'Ellsberg Urn' as one of their representations: subjects were told which objects are in the urn but were not told the quantities of each object so that the probability of drawing any particular object could not be known by the subject. Given that their objective is to examine the impact of different sources of ambiguity, Abdellaoui et al. (2009) [43] also considered other sources of ambiguity: changes in the French Stock Index, the temperature in Paris, and the temperature at some randomly drawn, remote country—all on a particular day. Halevy (2007) used the Ellsberg's Urn example as well [44].

---

[4] There is evidence (Speirs-Bridge et al., 2010 [37]) that people appear less confident when you give them an interval and ask them to estimate how likely it is that the correct answer is inside it, than if you specify a specific probability of being right and ask them for a confidence interval around it. In other words, people have higher confidence in probability estimates than confidence intervals.

Ahn et al. (2014)'s representation is simply not to tell subjects what the precise probability of two of the three possible outcomes was; this is a sort of a continuous Ellsberg Urn idea and inevitably suffers from the usual problem that the subjects may simply consider it the "suspicious urn" [45]. In contrast, Hey et al. (2010) [46] used an open and transparent representation: a Bingo Blower (see also footnote 7).

The papers by Hey et al. (2010) [46] and Abdellaoui et al. (2011) [9] used the traditional form of an experimental question, i.e., pairwise choice. On the contrary, Halevy (2007) presented reservation price questions [44], and Ahn et al. (2014) [45] used the allocation type of question that was pioneered originally by Loomes (1991) [47] but was forgotten for many years until revived by Andreoni and Miller (2002) [48] in a social choice context and later by Choi et al. (2007) [49] in a risky choice context.

Trautmann et al. (2013) [50] brought evidence that choice tasks elicit lower ambiguity aversion than valuation tasks (investment and certainty equivalent): in general, the type of elicitation task used has an important influence on measuring ambiguity attitudes (see also Maffioletti and al., 2009 [51]).

This paper reports the results of our attempt to develop a unique experimental framework aimed at disentangling the effects of internally-generated and externally-generated ambiguity situations on individual decision making in a setup, which controls for self-assessed levels of competence. We measure the ambiguity attitude by using both a 'willingness-to-bet' paradigm (choice between lotteries) and a 'willingness-to-invest' paradigm (investment choice) in order to control for framing effects and summarize the consequences of the two different sources—internal versus external—in two indexes obtained through a Principal Component Analysis. Our regression analysis shows that our measures of internal and external ambiguity aversion are significantly correlated, that the latter positively depends on performance, and the former depends positively on earnings and negatively on the perceived ease of the experiment.

In this paper, ambiguity is elicited by adopting a mixed approach that considers the contributions from both Abdellaoui (2009) [43] and Hey et al. (2010) [46] and uses the Bingo Blower as a representation of the traditional "Ellsberg Urn".

Our experimental design has a multistage structure that allows us to disentangle all the personal and contextual determinants of the decisions presented to experimental subjects (as discussed above) in order to evaluate their effect on individuals' ambiguity attitude.

Stage 1 is aimed at measuring individual ambiguity aversion through a single choice between ambiguity and a risky (50/50) setting.

Stage 2 focuses on individual ability in performance evaluation both in absolute and in relative terms (with respect to other subjects participating at the same session of the experiment) using two different questionnaires. The former (Questionnaire A) is chosen by the subject from among four questionnaires on different topics; the latter (Questionnaire B) is compulsory and equal for all participants. During this stage, subjects are asked to assess their ex-ante estimation of their ability in both in questionnaires and of their ex-ante perceived degree of competition in the selected Questionnaire A. Decisions were all incentivized.

Stage 3 is devoted to measurements of individual ex-post estimation of one's own ability and perceived ex-post placement in both questionnaires. Subjects' ambiguity attitudes are captured in an incentive-compatible way by asking subjects to bet on their estimations or choose between pairs of lotteries.

In Stage 4, the experimental tokens accumulated in the first three stages of the experiment constitute the endowment that subjects have the possibility of investing in two pairs of lotteries involving ambiguity and confidence[5]: after the investment decision, the computer randomly selects one of the two scenarios and plays out its consequences. This will determine the subject's individual payment from the experiment (plus the participation fee).

---

[5]　At the end of Stage 3 each subject faces a screenshot resuming all the outcomes of choices made in Stages 1 to 3 separately.

The experimental setting comprises two basic treatments (Self Selection and No Self Selection) each involving a one-shot game composed of four Stages played individually and separately by 15 subjects at the same time. The two treatments differ in the possibility of choosing the topic of Questionnaire A that occurs only in the Self Selection, giving subjects the chance to self-select in the preferred questionnaire; in the No Self Selection treatment, the computer randomly assigns the topic.

A more detailed description of tasks, decisions and incentives is reported in Section 4.

## 2. Results

### 2.1. Descriptive Statistics

#### 2.1.1. Attitude towards External Ambiguity

On average, subjects are indifferent between risk and external ambiguity based on the Bingo Blower: preference for BB-ambiguity is not significantly different from a preference for risk (*t*-test: $t = -0.316$, $p = 0.376$). For what concerns subjects' ability to estimate the content of the Bingo Blower correctly, 15% estimate the number of yellow balls correctly, while 10% are wrong (under/over-estimate it) by only 1 ball. Interestingly, when facing a non-incentivized choice between a risky urn and Ellsberg's urn, as in the final questionnaire, 72% of them prefer risk, 16% ambiguity, and 12% are indifferent.

#### 2.1.2. Over/Under-Estimation and Perceived Competence

For what concerns the choice of the specific Questionnaire A, subjects are quite homogeneously divided across the four topics: 25% of them chose Sport, 12% Showbiz, 31.5% History, and 31.5% Literature. Thus, the sample was well-calibrated in terms of ex-ante perceived competence.

The Questionnaire A subjects' estimation of their own performance is strongly influenced by self-perceived competence, both ex-ante and ex-post: although the average performance is significantly higher in Questionnaire B (10.14 vs. 11.93, *t*-test: $t = -4.317$, $p < 0.000$), before completing the two questionnaires subjects overestimate their score in Questionnaire A but underestimate their score in Questionnaire B (3.93 vs. $-0.92$, *t*-test: $t = 7.381$, $p < 0.000$). Miscalibration persists also in ex-post estimation after the effective engagement of subjects in the questionnaires, but the difference is reduced in entity and significance (0.78 vs. $-0.79$, *t*-test: $t = 2.178$, $p = 0.032$). The percentage of subjects who overestimate ex-ante their performance is 88% in Questionnaire A and 39% in Questionnaire B (*t*-test: $t = 7.532$, $p < 0.000$); the percentage of subjects who overestimate ex-post their performance is 47% in Questionnaire A and 30% in Questionnaire B (*t*-test: $t = 2.540$, $p = 0.013$). These results are summarized in Table 1.

**Table 1.** The estimation and perceived competence in the Self Selection treatment (averages).

| | Quest. A (High Competence) | Quest. B (Low Competence) | *p*-Value | Power |
|---|---|---|---|---|
| Ex-ante estimation of performance | 14.07 out of 20 | 11.01 out of 20 | $p < 0.000$ | 0.999 |
| Actual score | 10.14 out of 20 | 11.93 out of 20 | $p < 0.000$ | 0.819 |
| Ex-ante overestimation of performance | 3.93 | −0.92 | $p < 0.000$ | |
| % ex-ante overestimating subjects | 88% | 39% | $p < 0.000$ | |
| Ex-post estimation of performance | 10.22 out of 20 | 11.13 out of 20 | $p = 0.032$ | 0.619 |
| Ex-post overestimation of performance | 0.78 | -0.79 | $p = 0.032$ | |
| % ex-post overestimating subjects | 47% | 30% | $p = 0.013$ | |

Note: For all the items shown in the table, the sample size is equal to 89. Power is computed assuming alpha = 0.02.

### 2.1.3. Over/Under-Placement and Perceived Competence

Subjects' perceived placement is not affected by self-perceived competence, both ex-ante and ex-post. The percentage of subjects who declare ex-ante to be "above the average" is 48% in Questionnaire A and 51% in Questionnaire B (*t*-test: $t = 0.341$, $p = 0.733$), while the percentage of subjects who declare ex-post to be "above the average" is 10% in Questionnaire A and 16% in Questionnaire B (*t*-test: $t = -1.149$, $p = 0.253$). On average, subjects slightly overestimate the number of peers they are competing with in Questionnaire A. Table 2 summarizes these results.

**Table 2.** The placement and perceived competence in the Self Selection treatment (averages).

| | Quest. A (High Competence) | Quest. B (Low Competence) | *p*-Value |
|---|---|---|---|
| Overestimation of the degree of competition | 2.01 | | n.a. |
| % subjects who declare to be "above the average" (ex-post) | 48% | 51% | $p = 0.733$ |
| % ex-post overplaced subjects | 10% | 16% | $p = 0.253$ |

### 2.1.4. Over/Under-Precision and Perceived Competence When the Source of Ambiguity is "Internal"

Subjects' perceived competence affects subjects' propensity to bet on their own precision: 65% of subjects prefer to bet on their own precision in estimating their performance in Questionnaire A (instead of betting on a 50/50 risky lottery), whereas 55% of subjects prefer to bet on their own precision in estimating their performance in Questionnaire B (instead of betting on a 50/50 risky lottery) (*t*-test: $t = 1.450$, $p = 0.150$). Although this reveals no significant difference in terms of propensity to bet on precision, on average 56% of subjects are over-precise (i.e., wrongly bet on their own precision) in Questionnaire A, while 26% are over-precise in Questionnaire B (*t*-test: $t = 4.250$, $p = 0.0001$). It looks like feeling competent not only causes overconfidence in the sense of overestimation, but also makes people more prone to rely on their precision. This does not apply to placement. Preference for precision-based ambiguity is significantly higher than preference for risk in Questionnaire A (*t*-test: $t = 2.986$, $p = 0.002$), whereas it is not in Questionnaire B (*t*-test: $t = 0.953$, $p = 0.342$). See the summary of the results in Table 3.

**Table 3.** The precision and perceived competence in the Self Selection treatment.

| Quest. A (High Competence) | | Quest. B (Low Competence) | |
|---|---|---|---|
| # of subjects who bet on their precision | 58 out of 89 | # of subjects who bet on their precision | 49 out of 89 |
| % subjects who bet on their precision | 65% | % subjects who bet on their precision | 55% |
| # of subjects who wrongly bet on their precision (overprecise) | 50 out of 89 | # of subjects who wrongly bet on their precision (overprecise) | 23 out of 89 |
| % of subjects who wrongly bet on their precision (overprecise) | 56% | % of subjects who wrongly bet on their precision (overprecise) | 26% |

### 2.1.5. Over/Under-Precision and Perceived Competence when the Source of Ambiguity is "External"

When precision is not related to one's own performance in a task, but on the estimation of something that is unrelated to one's own performance (NY temperature, number of yellow balls in the Bingo Blower respectively), we obtain very similar results: in both cases preference for precision-based ambiguity is significantly lower than preference for risk (*t*-test: $t = -2.509$, $p = 0.007$ for both questions). In both tasks, 37% of subjects prefer to bet on their own precision (instead of betting on a 50/50 risky lottery). On average, 28% and 26% of subjects are over-precise (i.e., wrongly bet on their own precision) in the two tasks, respectively. This result corroborates the idea that the Bingo Blower provides a

good representation of external ambiguity. In sum, people are significantly less prone to rely on their precision when the estimation regards something unrelated to competence. See the summary of the results in Table 4.

**Table 4.** The precision and external ambiguity in the Self Selection treatment.

| Temperature of NY, 21 September (12 a.m.) | | Number of Yellow Balls in the Bingo Blower | |
|---|---|---|---|
| # of subjects who bet on their precision | 33 out of 89 | # of subjects who bet on their precision | 33 out of 89 |
| % subjects who bet on their precision | 37% | % subjects who bet on their precision | 37% |
| # of subjects who wrongly bet on their precision (overprecise) | 25 out of 89 | # of subjects who wrongly bet on their precision (overprecise) | 23 out of 89 |
| % of subjects who wrongly bet on their precision (overprecise) | 28% | % of subjects who wrongly bet on their precision (overprecise) | 26% |

### 2.1.6. Investment Framing

If we move from the pair-wise choice frame to the allocation or investment frame, we observe very similar findings. When facing Pair 1, subjects do not invest an average of 49% (665 out of 1337) of the tokens they have as their endowment; when facing Pair 2, subjects do not invest an average of 46% (621 out of 1337) of the tokens: there is no significant difference between the propensity not to invest across the two pairs of lottery (*t*-test: $t = 1.408$, $p = 0.162$). The two pairs both present a BB-based ambiguous lottery and differ because in Pair 1 the second lottery is an ambiguous one where ambiguity is related to performance in Questionnaire B, and in Pair 2 the second lottery is a 50/50 risky one. In Pair 1, subjects invest on average the 25% of tokens in the ambiguous lottery based on performance and the 28% in the BB-based ambiguous lottery; in Pair 2, subjects invest, on average, 25% of tokens in the ambiguous lottery based on performance and 26% in the BB-based ambiguous lottery. There is no significant difference in the number of tokens invested in the BB-based ambiguous lottery across the two pairs (*t*-test: $t = -1.050$, $p = 0.296$).

### 2.2. No Self Selection

The only difference between this treatment and the Self Selection concerns the fact that subjects cannot choose the topic of Questionnaire A but have to complete the Questionnaire the computer randomly assign them. The results show that the absence of self-selection has no significant effects on overestimation and overplacement, although the score in Questionnaire A is significantly lower in this treatment (8.90 out of 20 vs. 10.14, Mann–Whitney two-tailed test with $p = 0.04$, $z = -1.98$), emphasizing that subjects' choice of the topic in the Self Selection was "rational". Table 5 summarizes the results for the No Self Selection.

**Table 5.** The estimation and perceived competence in the No Self Selection treatment (averages).

| | Quest. A (High Competence) | Quest. B (Low Competence) | *p*-Value | Power |
|---|---|---|---|---|
| Ex-ante estimation of performance | 12.43 out of 20 | 9.97 out of 20 | $p = 0.014$ | 0.859 |
| Actual score | 8.90 out of 20 | 10.93 out of 20 | $p < 0.000$ | 0.820 |
| Ex-ante overestimation of performance | 3.52 | −0.95 | $p < 0.000$ | |
| % ex-ante overestimating subjects | 64% | 43% | $p < 0.000$ | |
| Ex-post estimation of performance | 9.84 out of 20 | 10.61 out of 20 | $p = 0.074$ | 0.673 |
| Ex-post overestimation of performance | 0.93 | −0.31 | $p = 0.032$ | |
| % ex-post overestimating subjects | 48% | 41% | $p = 0.017$ | |

Note: For all the items shown in the table, the sample size is equal to 74. Power is computed assuming alpha = 0.02.

For what concerns overestimation ex-ante, on average subjects in this treatment tend to overestimate their score in Questionnaire A by 3.52 correct answers vs. 3.93 in the Self Selection ex-ante: the difference is not significant between the two treatments (Mann-Whitney two-tailed test with $p = 0.42$, $z = -0.79$). The same holds for overestimation ex-post: on average, subjects tend to overestimate their score of 0.93 correct answers vs. 0.08 in the Self Selection (Mann-Whitney two-tailed test with $p = 0.41$, $z = 0.82$).

Regarding placement, no significant difference emerges as well: 50% of subjects overplace themselves vs. 48% in the Self Selection (Mann-Whitney one-tailed test with $p = 0.43$, $z = -0.18$) (see Table 6).

**Table 6.** The placement and perceived competence in the No Self Selection treatment.

| | Quest. A (High Competence) | Quest. B (Low Competence) | *p*-Value | Power |
|---|---|---|---|---|
| Overestimation of the degree of competition (average) | 2.20 | | n.a. | |
| % subjects who declare to be "above the average" (ex-post) | 50% | 50% | $p = 1.000$ | 0.080 |
| % ex-post overplaced subjects | 14% | 7% | $p = 0.322$ | 0.089 |

Note: For all the items shown in the table, the sample size is equal to 74. Power is computed assuming alpha = 0.02.

### 2.3. Principal Component Analysis

Principal component analysis (henceforth PCA) has been increasingly used for the creation of indexes of social economic status (Kolenikov and Angeles., 2009 [52]). To the best of our knowledge, this technique has never been employed for creating indexes of uncertainty or ambiguity attitude, but we believe it can be successfully applied to synthesize our data. The reason is that our experiments provide different measures of ambiguity, potentially correlated, some of which may yield similar information, with no measure that can be judged a priori as better than the others to capture subjects' ambiguity attitude. PCA is a multivariate statistical technique used to reduce the number of variables in a data set into a smaller number of "dimensions": in mathematical terms, from an initial set of $n$ correlated variables, PCA creates uncorrelated indexes or components, where each component is a linear weighted combination of the initial variables (Jolliffe, 2002 [53]). Only binary variables can be used, so we restrict PCA to measures of ambiguity attitude based on the choice between an ambiguous lottery (the variable assumes the value of zero) and a risky one (the variable assumes the value of one). Thus, for what concerns external ambiguity, only ambiguity measured within the 'willingness-to-bet' paradigm is considered. We also recode the self-reported hypothetical choice between a fully-ambiguous urn and a risky one as binary. PCA works best when asset variables are correlated, but also when the distribution of variables varies across cases, in this case, subjects. Thus, a natural approach is to use methods such as PCA to try and organize the data to reduce its dimensionality with as little loss of information as possible in the total variation these variables explain (Giri, 2004 [54]). The output from a PCA is a table of factor scores or weights for each variable (see Table 7). In our setting, a variable with a positive factor score is associated with higher ambiguity propensity, and conversely, a variable with a negative factor score is associated with lower ambiguity propensity. Data from both treatments are pooled together.

Results from the first principal component for our sample of 133 subjects are shown in Table 7 and their associated eigenvalues are 1.22 (internal ambiguity), and 1.15 (external ambiguity), accounting for 30.07% and 38.17%, respectively, of the variation in the original data.

It is the ambiguity measures that vary more across subjects that are given more weight in PCA (McKenzie 2003 [55]): for example, if all subjects choose the same lottery in a pair (i.e., zero standard deviation) it would exhibit no variation between subjects and would be zero weighted, and so of little use in differentiating ambiguity. For what concerns internal ambiguity, the measure that carries higher weight is the preference for high-competence based ambiguity; the preference for investing in one's

own competence in counting Bingo Blower balls shows a high weight, too; the lower weight is the propensity to invest in the lottery where the probability of winning is based on relative performance. Vice versa, regarding external ambiguity, the propensity to invest in the Bingo Blower lottery is the measure that counts more, followed by the propensity to bet on the Bingo Blower; betting on an item on which subjects have almost no clue (NY temperature) has a very low weight. These results are in line with preference reversal from lottery-based measures to investment-based measures shown in the literature: what is interesting here is that the different types of measures imply different weights, depending on whether ambiguity is internal vs. external.

**Table 7.** The Principal Component Analysis.

| Variable Description | Mean | Std. Deviation | Factor Score |
|---|---|---|---|
| **Internal Ambiguity** | | | |
| Preference for high-competence-based ambiguity (vs. risk) | 0.684 | 0.466 | 0.684 |
| Preference for low-competence-based ambiguity (vs. risk) | 0.526 | 0.501 | 0.274 |
| Preference for no-competence-based ambiguity (BB # yellow balls) (vs. risk) | 0.360 | 0.402 | 0.672 |
| Investment in competence-based-ambiguity (vs. risk) | 0.541 | 0.500 | 0.067 |
| **External Ambiguity** | | | |
| Preference for BB-based ambiguity (vs. risk) | 0.511 | 0.501 | 0.703 |
| Preference for no-competence-based ambiguity (temperature) (vs. risk) | 0.375 | 0.486 | 0.044 |
| Investment in BB-ambiguity (vs. risk) | 0.684 | 0.466 | 0.709 |

Note: "High-competence-based ambiguity" refers to choices related to subjects' performance in Questionnaire A. "Low-competence-based ambiguity" refers to choices related to subjects' performance in Questionnaire B.

Using the factor scores from the first principal component as weights, for each subject a dependent variable for each type of ambiguity—which has a mean equal to zero, and a standard deviation equal to one—can then be constructed. This dependent variable can be interpreted as follows. The former is the subject "internal ambiguity score": the higher the subject's ambiguity score, the higher the implied ambiguity propensity related to the confidence of that subject. The latter is the subject "external ambiguity score": the higher the subject's ambiguity score, the higher the implied ambiguity propensity related to the context of that subject. Since the correlation between the two scores is unknown a priori, we run two distinct regressions and simultaneously test for correlation between the two dependent variables. Table 8 summarizes the Seemingly Unrelated Regression (SUR) model results.

First, the regression shows that subjects with higher external ambiguity scores show higher internal ambiguity scores, and vice versa: the correlation coefficient is equal to 0.212 and is significant at 1.4% level. Thus, it appears that subjects who are disturbed by ambiguity derived from internal sources (under-confident subjects) are significantly affected by ambiguity originating from external sources. Self-selection plays no role, as anticipated by the descriptive statistics presented above. Furthermore, highly-skilled subjects look more ambiguity-averse for what concerns internal ambiguity, suggesting that smarter people are able to evaluate themselves and are rational in refusing ambiguity. External ambiguity propensity is slightly affected by earnings (negatively) and by the perceived ease of the experiment (positively): subjects who earn more are less ambiguity-seeking and subjects who evaluated the experiment as "easy" are more ambiguity-seeking.

**Table 8.** The determinants of Internal and External Ambiguity Scores.

| SUR Model | | | |
|---|---|---|---|
| **Internal Ambiguity Score** | | **External Ambiguity Score** | |
| Treatment | 0.094 | Treatment | −0.356 |
| | (0.226) | | (0.217) |
| Score high-competence task | 0.031 ** | Score high-competence task | 0.044 |
| | (0.031) | | (0.029) |
| Score low-competence task | 0.025 | Score low-competence task | 0.014 |
| | (0.037) | | (0.036) |
| Earnings | 0.000 | Earnings | −0.001 * |
| | (0.001) | | (0.001) |
| Gender | 0.013 | Gender | −0.071 |
| | (0.184) | | (0.191) |
| Age | −0.005 | Age | 0.012 |
| | (0.044) | | (0.046) |
| Easy | 0.268 | Easy | 0.013 * |
| | (0.190) | | (0.183) |
| Constant | −0.979 | Constant | 0.235 |
| | (1.262) | | (1.216) |

Correlation of residuals: 0.212. Breusch-Pagan test of independence: chi2 (1) = 5.979, Pr = 0.0145. Internal Ambiguity Score: $R^2$ = 0.269, External Ambiguity Score: $R^2$ = 0.274. The dependent variables range from 0 to 1. Controls: geographical origin, past involvement in previous experiments, the main motivation driving the subjects' choices during the experiment. ** significant at 5%; * significant at 10%.

## 3. Discussion

This paper provides a novel methodological contribution to ambiguity measurements when ambiguity is derived from internal or external sources. Using Principal Component Analysis, the subjects' choices are translated into two indexes or measures of internal and external ambiguity attitudes that are shown to be positively and significantly correlated: more confident subjects are also the subjects that are more able to tolerate ambiguity originating from the context. Interestingly, our measure of internal ambiguity depends negatively on performance in the high-competence questionnaire, suggesting that more skilled individuals are more ambiguity-averse (or less confident) when ambiguity derives from an internal source. The results are not driven by self-selection: the possibility of choosing the topic of the high-competence task affects subjects' performance positively but has no effect on confidence. The analysis of external ambiguity determinants shows two small-sized but significant effects: the higher the subject's earnings, the higher the ambiguity-aversion; subjects who perceived the whole experiment as "easy" are significantly less ambiguity-averse.

## 4. Materials and Methods

Subjects were recruited via ORSEE (Greiner, 2004 [56]). We ran nine computerized sessions using the z-Tree software (Fischbacher, 2007 [57]) between May 2015 and September 2015, with a total amount of 133 participants (89 subjects in the Self Selection treatment and 44 subjects in the No Self Selection treatment). Participants were undergraduate students (48% males). We employed a between-subjects design: no individual participated in more than one session. In each session, the participants were paid a 5€ show-up fee, plus their earnings from the experiment (with average earnings equal to 11.92€). At the beginning of each session, participants were welcomed and, once all of them were seated, the instructions of all stages were handed to them in written form before being read aloud by one

experimenter. The instructions of each stage were also presented on subjects' computer screen at the beginning of the stage. More than sixty percent of subjects classified the experiment as "easy"[6]; sessions took approximately one hour.

### 4.1. Self Selection Treatment

*Stage 1.* Subjects have to choose between two lotteries: the ambiguous lottery X where they can win $W_1$ if a yellow ball is drawn from the Bingo Blower or 0 if a pink or blue ball is drawn, and risky lottery Y, where they win the same amount, $W_1$, with a (known) probability of $\frac{1}{2}$ and 0 with probability $\frac{1}{2}$. The Bingo Blower is located in the room where the experiment takes place and contains yellow, pink and blue balls. This choice is meant to measure *subjects' preference for BB-ambiguity* ("BB" refers the Bingo Blower since ambiguity is measured here by means of the Bingo Blower[7]) vs. *risk*.

*Stage 2.* Subjects will face two questionnaires or tasks: Questionnaire A is based on a specific skill or knowledge on a topic they can choose from among four options (sport, showbiz, history, and literature), and Questionnaire B is a general knowledge task that is compulsory and equal for everybody. Both questionnaires involve multiple-choice questions with four possible answers, where only one is correct. To ensure subjects put proper effort into picking the correct answer, both questionnaires are monetarily incentivized: they will earn a certain amount of tokens for each correct answer. Since they are supposed to select Questionnaire A on the basis of their competence, we can consider Questionnaire A as a "high-competence" (at least in subjects' perception) task and Questionnaire B as a "low-competence" task. We elicit subjects' beliefs about their competence in answering both the questionnaires (on a 0–20 scale) before they face them. This answer captures subjects' ex-ante self-evaluation of competence in absolute terms (*ex-ante estimation* in the high-competence task and in the low-competence task). Furthermore, for Questionnaire A subjects have to guess how many participants in the session chose the same questionnaire they did. This is to identify the number of subjects they expect to compete with (*perceived degree of competition*). For both (incentivized) guesses, they earn $W_2$ and $W_3$ tokens respectively if they are correct or if they over/underestimate the correct number by 1 unit.

*Stage 3.* First, subjects have to evaluate their absolute performance in Questionnaire A (*ex-post estimation* in the high-competence task) they selected: the closer to the effective score Questionnaire B (*ex-post estimation)* their prediction is, the higher number of tickets they receive for taking part in a lottery called Alpha. Subjects can choose between playing lottery Alpha—where all the tickets earned by the participants in the session are pulled together, only one wins $W_4$ and the others get zero—or another lottery (Beta) where they win either $W_4$ or zero with probability *1/n*, where *n* is the number of participants in the session. This choice identifies *subjects' preference for internal ambiguity (based on relative precision,* i.e., a measure of overprecision) in the high-competence task with respect to risk.

Second, subjects have to evaluate their absolute performance in the low-competence task and choose between betting on the correctness of their answer (they win if they over/underestimate by one correct answer at the maximum) or on a 50/50 lottery. In both lotteries, they win $W_5$ or get zero. This choice identifies subjects' preference for internal ambiguity (based on placement, i.e., a measure of overplacement) in the low-competence task with respect to risk.

---

[6] Although the experiment seems at first sight quite cumbersome, this result confirmed that participants were able to understand it; this is also indirectly confirmed by the fact that all sessions lasted one hour and half without anybody delating the others.

[7] The Bingo Blower is a rectangular-shaped, glass-sided: inside the glass walls are a set of pink, yellow, and blue balls in continuous motion being moved about by a jet of wind from a fan in the base (see the picture in Appendix A). In addition, during the sessions' images of the Bingo Blower in action are projected via videocamera onto two big screens in the laboratory. Subjects are free at any stage to get closer to the Blower to examine it as much as they want. All the balls inside the Blower can at all times be seen by people outside, but, unless the number of balls in the Blower is low, the number of balls of differing colors cannot be counted because they are continually moving around: while objective probabilities do exist, subjects cannot know them. In this way, as noted by Hey and Pace (2014) [58], there is a "situation of genuine ambiguity which eliminates the problem of suspicion; the problem of using directly a second-order probability distribution; and the problem of using real events, therefore keeping the problem more similar to the original Ellsberg one".

Third, subjects have to guess whether their score in task B is higher, equal or lower than the average score in the session. They earn $W_6$ tokens if they are correct. This guess identifies *subjects' ex-post placement* in the low-competence task.

Fourth, subjects have to estimate the temperature in New York City on a specific past date and time (17 September 2014, at noon) and choose between betting on the correctness of their answer (they win if they over/underestimate by one degree Celsius at the maximum) or on a 50/50 lottery. In both lotteries, they win $W_7$ or get zero. This choice captures *subjects' preference for external ambiguity* (based on an exogenous source, i.e., a measure of overestimation) with respect to risk. Furthermore, subjects have to guess whether their estimation of NYC temperature is more, equally or less correct than the average temperature estimated in the session. This guess identifies subjects' ex-ante estimation in case of external ambiguity.

Fifth, subjects have to estimate the number of yellow balls in the Bingo Blower and choose between betting on the correctness of their answer (they win if they over/underestimate by one ball at the maximum) or on a 50/50 lottery. In both lotteries, they again win $W_7$ or get zero. This choice captures another form of subjects' *preference for external ambiguity* based on an exogenous source, the Bingo Blower, with respect to risk. This time, however, subjects could feel to have some kind of "control" and/or direct experience with the source of external ambiguity since the Bingo Blower was located in the room where the experimental sessions took place and subjects could observe it as long as they wanted and get as close to it as they liked in order to increase their perceived accuracy in estimating the number of yellow balls. This guess identifies subjects' *ex-ante estimation in case of external ambiguity*. Furthermore, subjects have to guess whether their estimation of the number of yellow balls is more, equally or less correct than the average number estimated in the session.

*Stage 4.* Subjects receive feedback on their total amount of earnings and decide how many tokens—of the sum earned in the previous stages and that we call endowment $D_i$, different for each subject $i$—they want to allocate between two pairs of lotteries ("Pair 1" and "Pair 2" respectively): they will decide the allocation of the tokens they earned for both pairs, but only one pair will be selected at random and actually played. The structure of this "investing gamble" is grounded on Gneezy and Potters (1997) [59].

Pair 1:

- Investment Choice 1a: subjects have to decide how many tokens ($G_i$) out of their endowment $D_i$ that they want to allocate to a lottery where the probability of winning 2.5 $G_i$ (instead of getting zero) corresponds to the percentile corresponding to their performance in task B.

- Investment Choice 1b: subjects have to decide how many tokens ($H_i$) out of their endowment $D_i$ that they want to allocate to a lottery where they win 2.5 $H_i$ (instead of getting zero) if a yellow ball is drawn from the Bingo Blower.

Obviously, the condition $G_i + H_i \leq D_i$ must hold.

The ratio $G_i/H_i$ in Pair 1 captures *subjects' preference for investing in internal ambiguity* (based on placement) in the low-competence task instead of investing in external BB-ambiguity.

Pair 2:

- Investment Choice 2a: subjects have to decide how many tokens ($G_i$) out of their endowment $D_i$ that they want to allocate to a lottery where the probability of winning 2.5 $G_i$ is $\frac{1}{2}$ and the probability of getting 0 is $\frac{1}{2}$.

- Investment Choice 2b: subjects have to decide how many tokens ($H_i$) out of their endowment $D_i$ that they want to allocate to a lottery where they win 2.5 $H_i$ (instead of getting zero) if a yellow ball is drawn from the Bingo Blower.

Again, the condition $G_i + H_i \leq D_i$ must hold.

The ratio $G_i/H_i$ in Pair 2 captures subjects' preference for investing in external BB-ambiguity vs. risk.

Finally, subjects have to estimate the number of yellow balls in the bingo-blower. They win $W_8$ if they over/underestimate by one ball at the maximum. This is to capture how correct subjects are when

evaluating external BB-ambiguity (*BB-calibration*) and serves as additional information to disentangle between internal and external BB-ambiguity.

After the end of the experiment, subjects have to answer a final set of questions aimed at collecting demographic information (gender, age, geographic origin, experience in taking part in experiments, perceived difficulty of the present experiment) plus a question on the motivation of their choices in the experiment ("maximize private earnings", "being in line with other participants", "altruism") and a hypothetical question asking whether they preferred to bet on an urn of unknown composition or on a 50/50 risky urn, where subjects could also indicate indifference between the two urns.

Regarding our payment protocol, in Stages 1–3, the subjects' payment depends on the outcome of each lottery they decide to face. Thus, each chosen option is paid before a subsequent decision is made. For what concerns Stage 4 only, subjects have to make decisions regarding ten scenarios, and only one of them is selected at random and actually implemented. According to Cox et al. (2015)'s classification [60], our payment protocol refers to the case where the experimenter "pays all sequentially" in Stages 1–3, and "randomly pays one decision for each subject" in Stage 4 [60]. Subjects' revealed risk preferences are shown in Cox et al. (2015)'s paper to differ across mechanisms. Nonetheless, both the mechanisms we chose are demonstrated to be incentive compatible [60].

### 4.2. No Self Selection Treatment

This treatment differs from the Self Selection just in the fact that subjects do not choose the specific Questionnaire A (on the topic they feel more competent in), but the topic of Questionnaire A is randomly assigned by the computer. In Stage 2, the specific Questionnaire A they have to complete is randomly assigned by the computer: the four possible topics are the same as the Self Selection and the same probability can be drawn. This treatment works as a control for the effects of self-selection into the task subjects feel competent in and helps us disambiguate between the role of perceived competence and the role of self-selection.

The instructions and the final questionnaire are reported in Appendix B.

## 5. Conclusions

The literature review above acknowledges the existence of some studies on overconfidence and risk aversion, but we are aware of no study that focuses on this specific, although very important issue, namely the relationship between overconfidence and ambiguity aversion. Furthermore, we are able to measure the three types of overconfidence (i.e., overestimation, overplacement and overprecision) in the same experiment, which is something that none of the above-cited papers does in decisions under risk.

**Author Contributions:** Conceptualization, D.D.C. and D.G.; Formal analysis, D.D.C. and D.G.; Funding acquisition, D.D.C. and D.G.; Methodology, D.D.C. and D.G.; Writing—original draft, D.D.C. and D.G.

**Funding:** This research was funded by LUISS (to D.D.C.) and by Fondation du Risque, Université Paris-Dauphine, ENSAE and Groupama under the aegis of the Research Chair entitled "Individuals and Risk: analysis and market response" (to D.G.).

**Acknowledgments:** We thank Andrea Lombardo for valuable technical support with the experimental software. We are grateful to Werner Guth and John Hey and to the participants of the ESA 2014 Conference in Prague, the LABSI 2014 Workshop in Siena and LUISS internal seminar for very helpful comments and suggestions. All errors remain ours.

**Conflicts of Interest:** The authors declare no conflict of interest.

## Appendix A

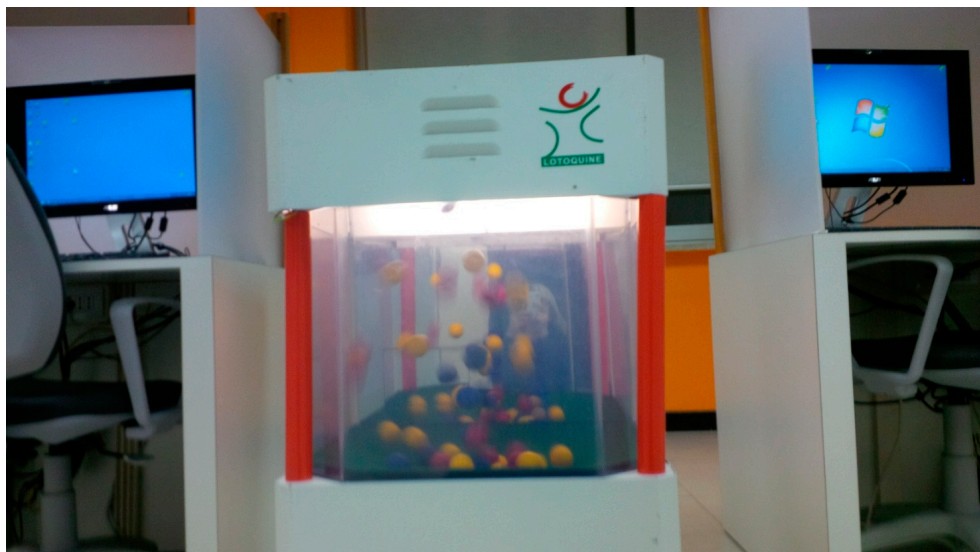

**Figure A1.** The Bingo Blower (photo taken in the CESARE experimental lab).

## Appendix B

### Instructions

Welcome to our experiment!

Please read the Instructions carefully.

You are taking part in an experiment in which you will make several economic investment decisions. These decisions and chance will determine your payment from the experiment, which you will be paid privately in cash immediately at the end of the session.

Your decisions will remain completely anonymous: this means that the experimenters will not be able to associate any of them to your name. All decisions and answers will be stored using an anonymous identifier rather than any personal information.

During the experiment, we ask that you work individually and that you do not communicate with other participants. If you do not follow this rule you may be excluded from the experiment and forego your earnings. Please pay attention to the instructions that will be presented on your screen, which will also be read aloud by the experimenters.

Once the computer tasks have ended, you will be asked to answer a brief questionnaire.

This experiment consists of 4 different Stages.

- In Stage 1 you will be asked to make a simple choice between two lotteries.
- In Stage 2 you will answer questions presented in two different Questionnaires with 20 questions each, and you will be asked to make an evaluative assessment.
- In Stage 3 you will be asked to make some forecasts and on this basis to choose between 4 related pairs of lotteries that will be presented in sequence to you.

Your decisions and the subsequent outcomes in Stages 1, 2 and 3 will allow you to accumulate experimental token earnings that you will have the possibility to use in Stage 4.

- In Stage 4 you will have the possibility to allocate the tokens you accumulated in the previous Stages between two different investment opportunities.

You will receive detailed instructions of each Stage of the experiment before they start.

Please work in silence and do not disturb other participants. If you have some questions raise your hand and wait: one experimenter will come and help you as soon as they can.

Enjoy.

**Stage 1**

You can see on the big screen situated in front of you the BINGO BLOWER, where red balls, yellow balls and blue balls are in movement.

Choose which of the two following lotteries you prefer to play:

LOTTERY X: in which you win 50 tokens if a yellow ball is randomly drawn from the Bingo Blower and nothing otherwise.

LOTTERY Y: in which you win 50 tokens if an even number results from a roll of a dice and nothing otherwise.

**Stage 2**

During this Stage, you will be asked to answer the questions included in two different questionnaires (namely "Questionnaire A" and "Questionnaire B"), each composed of 20 multiple choice questions, and then to make some evaluation about them.

Questionnaire A offers questions on four different topics: SPORTS (Questionnaire A1), ENTERTAINMENT (Questionnaire A2), HISTORY (Questionnaire A3), and LITERATURE (Questionnaire A4). You may choose one of these four questionnaires to respond to.

Questionnaire B is the same for all participants in the session and includes General Culture questions.

All Questionnaires are composed of 20 multiple choice questions. Each question has four possible answers, among which there is only one correct answer. For each correct answer, you will gain 40 experimental tokens.

There is a time limit of 25 s for answering each question; if you do not answer a question within this time limit, the computer will automatically move to the next one. You earn nothing from any questions that you fail to respond to in this way.

Stage 2 is composed of several STEPS.

In STEP 1 the computer will ask you to decide which Questionnaire A (A1, A2, A3, A4) you are willing to answer.

In STEP 2, before answering the chosen Questionnaire A, the computer will ask you to declare how confident you feel about your ability to answer Questionnaire A correctly (on a scale basis from 0 to 20). If your evaluation of how many questions you will get correct is fulfilled (i.e., the number of your correct answers to Questionnaire A is equal to your evaluation (+ or −1)) you will receive 50 experimental tokens.

In STEP 3 you will be asked to guess how many participants in your session have chosen the same Questionnaire selected by you. If you are able to guess it (+ or −1), you will receive 50 experimental tokens.

In STEP 4 you will answer the Questionnaire A that you chose.

In STEP 5, before answering Questionnaire B, the computer will ask you to declare how confident you are about your ability to answer it correctly (on a scale basis from 0 to 20). If you are correct about how many answers you will get correct on Questionnaire B (+ or −1)) you will receive 50 experimental tokens.

In STEP 6 you will answer Questionnaire B.

**Stage 3**

Stage 3 is composed of 4 Rounds. In each Round, you will state your predictions about different situations and then you will face a pair of lotteries that you will be asked to choose between. Each round is composed of 3 Steps.

**Round 1**

In Step 1.1—you will again be asked to predict how many correct answers you got in Questionnaire A chosen in Stage 2 (prediction 1.1); the more accurate your prediction (i.e., the closer your prediction is to the actual correct number you got) the higher your probability of winning in the Lottery "ALFA", explained below.

In Step 1.2—on the basis of your prediction and without having any feedback on your actual score, you will be asked to choose between:

LOTTERY "ALFA"—in which you win 200 tokens with a probability that is determined in the way shown in the table on your computer screen (which links your prediction to your possible actual performance on Questionnaire A to your chances of winning) and nothing otherwise;

and

LOTTERY "BETA"—in which you win 200 tokens with a chance of 50% and nothing otherwise.

In Step 1.3—you will be asked to predict by how much your score in the chosen Questionnaire A is better or worse than that of the other participants present here today who selected the same Questionnaire (i.e., whether your number of correct answers is greater than or less than the average of their correct answers). If your prediction is correct (+ or −1) you will receive 50 experimental tokens.

**Round 2**

In Step 2.1—you will be asked to predict how many correct answers you got in Questionnaire B (the questionnaire that is the same for all participants) (prediction 2.1); the more accurate your prediction is (i.e., the closer your prediction is to the actual correct number you got) the higher your probability of winning in the Lottery "GAMMA".

In Step 2.2—on the basis of your prediction and without having any feedback on your actual score, you will be asked to choose between:

LOTTERY "GAMMA"—in which you win 200 tokens with a probability that is determined in the way shown in the table on your computer screen (which links your prediction to your possible actual performance on Questionnaire B to your chance of winning) and nothing otherwise;

and

LOTTERY "DELTA"—in which you win 200 tokens with a chance of 50% and nothing otherwise.

In Step 1.3—you will be asked to predict by how much your score in Questionnaire B is better or worse than that of other participants present here today (i.e., whether your number of correct answers is greater than or less than the average of their correct answers). If your prediction is correct (+ or −1) you will receive 50 experimental tokens.

**Round 3**

In Step 3.1—you will be asked to predict the temperature registered in New York City at 12.00 p.m. on 17 September 2014 (prediction 3.1); the more accurate your prediction is, the higher your probability of winning in the Lottery "EPSILON".

In Step 3.2—on the basis of your forecast and without having any feedback on your actual score, you will be asked to choose between:

LOTTERY "EPSILON"—in which you win 200 tokens with a probability that is determined in the way shown in the table on your computer screen (which links your prediction accuracy of the temperature in New York to your chance of winning) and nothing otherwise;

and

LOTTERY "ZETA"—in which you win 200 tokens with a chance of 50% and nothing otherwise.

In Step 3.3—you will be asked to predict if your prediction of the temperature in New York is better or worse than the average prediction of the participants in your session. If your forecast is correct, you will receive 50 experimental tokens.

**Round 4**

In Step 4.1—you will be asked to guess how many yellow balls are in the Bingo Blower (prediction 4.1); the more accurate your prediction is, the higher your probability of winning in the Lottery "ETA".

In Step 4.2—on the basis of your prediction and without having any feedback on your actual score, you will be asked to choose between:

LOTTERY "ETA"—in which you win 200 tokens with a probability that is determined in the way shown in the table on your computer screen (which links your prediction accuracy of the balls in the Bingo Blower to your chance of winning) and nothing otherwise;

and

LOTTERY "IOTA"—in which you win 200 tokens with a chance of 50% and nothing otherwise.

In Step 4.3—you will be asked to predict if your prediction of the number of yellow balls in the Bingo Blower is better or worse than the average prediction of the participants in your session. If your prediction is correct, you will receive 50 experimental tokens.

**Stage 4**

In Stage 4 you will receive detailed information on how many experimental tokens you gained from the previous STAGES of the experiment (namely STAGE 1, 2 and 3). We will call this gain your Endowment (D).

The computer will then ask you to decide how much (if any) of that Endowment (D) you are willing to invest in the following two separate and independent investment opportunities.

**Warning: In each of these you should decide how to allocate your entire endowment. Only one of these will actually be carried out, which will be determined randomly on the computer.**

*Each investment opportunity consists of a pair of options for you to choose between. Be careful to decide in each pair which one you prefer because at the end of the session the computer will randomly select one of the pairs (with a probability of 50%) and will carry out the opportunity of investment you preferred in it for real payment.*

**Pair 1**

**Investment Opportunity 1a**—you invest G tokens from your endowment D, where G is the number of tokens of your choice: this will give you the chance to win 2.5* G with a probability corresponding to your ranking in Questionnaire B. For example, if your ranking in Questionnaire B implies that you did better than 80% of participants in your session, your probability of winning will be 80%, if you did better than 20% of participants, so that 80% of the participants did better than you, your probability of winning will be 20%.

**Investment Opportunity 1b**—you invest H tokens from your endowment D, where H is the number of tokens of your choice: this will give you the possibility of winning 2.5* H if a yellow ball is drawn and nothing if it is a blue or red ball.

*Importantly, also notice that you can decide not to invest any of your tokens in any of the two investment opportunities or you can decide to invest all of them in one or the other or you can decide to invest some tokens in one opportunity and some in the other (and perhaps also keep some without investing them). Thus, any division of the tokens between opportunity 1a, opportunity 1b, and not investing is allowed. The only rule that you have to respect is that the sum of tokens invested overall (G + H) does not exceed your endowment (D): G + H ≤ D.*

**Pair 2**

**Investment Opportunity 2a**—you invest G tokens from your endowment D: this will give you the possibility of winning 2.5* G with a probability of 50% and nothing otherwise;

**Investment Opportunity 2b**—you invest H tokens from your endowment D: this will give you the possibility of winning 2.5* H if a yellow ball will be drawn out and nothing if it is a blue or red ball.

*As for Pair 1, you can choose any division of your tokens that you like between Investment Opportunity 2a, Investment Opportunity 2b, and keeping them, i.e., not investing. The only rule that you have to respect is that the sum of tokens invested overall (G + H) does not exceed your endowment (D): G + H ≤ D.*

**FINAL QUESTIONNAIRE**

(1) Age
(2) Sex
(3) Year of course
(4) Geographical origin
- North
- Center
- South

(5) Did you find the experiment easy?
- Yes
- No

(6) How many experiments have you attended up to today?
- From 1 to 5
- More than 5

(7) When you have an important task to complete, do you care more about your performance or about how you perform in comparison with the others that are making the same task?
- My score
- Performing in line with the others *(if this option selected, answer classified as "altruism")*
- Both

(8) Imagine two urns (A and B). Urn A contains 100 balls, 50 black and 50 white; Urn B contains 100 balls either all black or all white. You know that if you draw a white ball you win. From which urn will you draw your ball?
- Urn A (if this option is selected, the answer is classified as "preference for the risky urn")
- Urn B (if this option is selected, the answer is classified as "preference for the Ellsberg urn")
- At random

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
