# Peer review of "Measuring and Disentangling Ambiguity and Confidence in the Lab"

_games, doi:10.3390/g10010009_

Round 1
Reviewer 1 Report
Report on the paper:
Measuring and Disentangling Ambiguity and Confidence in the Lab
Summary
This study is very rich in both the research questions and methodology. It addresses experimentally the issue of ambiguity and confidence jointly within a unique experimental framework. In particular, it aims at disentangling the effects of internally-generated and externally-generated ambiguity situations on individual decision making, by controlling for self-assessed levels of competence.
As for confidence, the experimental design allows the authors to measure the three types of overconfidence introduced by Moore and Healy (2008) – overestimation, overplacement and overprecision. As for ambiguity, subjects’ ambiguity attitude is measured by using both a ‘willingness-to-bet’ paradigm (choice between lotteries) and a ‘willingness-to-invest’ paradigm (investment choice) in order to control for framing effects.
A Principal Component Analysis is performed to examine the behavioral consequences of internallygenerated and externally-generated ambiguity. The regression analysis shows that the authors’ measures of internal and external ambiguity aversion are significantly correlated. However, while the latter positively depends on performance, the former positively depends on earnings and negatively depends on the perceived ease of the experiment.
General Statement:
I think that this paper represents a fine addition to the experimental literature on ambiguity aversion and to the one on (over-)confidence, thereby deserving to be published in Games.
I only have a list of minor comments that I ask the authors to implement in the next version of the paper. The majority of these comments concern typos, misleading sentences, or bad organization of this version of the paper. Some others regard additional studies that I think should be cited and discussed, since they are relevant to the authors’ research questions and methodology.
In the minor comments below, I refer to the authors as “you”.
Comments:
Page 1, line 34. Replace “knowledge,e” with “knowledge,”
Page 1, lines 36-42. You say that very few studies address the issues of ambiguity and confidence jointly, and you only cite Brenner et al. (2011). I am aware of three other studies that contribute to this specific field: Gysler, Brown Kruse & Schubert (2002), Shyti (2013), and Yang & Zhu (2016). I acknowledge that two of them are unpublished and that they all focus on financial or entrepreneurial decisions. Despite that, I think that you might quickly discuss in which regards these three studies relate or not to your work.
Page 2, lines 52-54. The way you relate ambiguity to confidence, besides being coherent with Abdellaoui et al. (2011) source of uncertainty, is also coherent with Attanasi, Gollier, Montesano & Pace (2014) belief elicitation under ambiguity. The latter generate ambiguity in a two-stage setting à la Klibanoff, Marinacci & Mukerji (2005) by relying on different “external” sources in the first stage (i.e., binomial distribution, uniform distribution, unknown distribution) at a between-subject level. Each external source is generated by a common mechanism of uncertainty at a within-subject level. Then, given the second-stage urn composition (which is unknown), they elicit subjects’ beliefs about it. The heterogeneous distribution of beliefs they obtain can be interpreted as an “internal” source of ambiguity, i.e., in your language, different levels of “confidence” for different subjects. They find no correlation between this “confidence” and subjects’ behavior under ambiguity. I think that reporting also this study would strengthen the relation between your study and the previous experimental literature.
Page 2, lines 64-67. I agree with you that the effects of overconfidence (i.e., overestimation, overplacement or overprecision) on ambiguity attitudes has never been previously studied in the literature (to the best of my knowledge). However, I am aware of two articles that have studied the effects of overconfidence on risk attitudes. Both these studies focus on voting choice, modeling it as a risky lottery. The first one – Attanasi, Corazzini, Georgantzis & Passarelli (2014) – studies subjects’ overconfidence and its relation with risk aversion in a theory-driven experiment where each subject can get an informative signal about the others’ voting preferences. The second one – Attanasi, Corazzini & Passarelli (2017) – explicitly refers to Moore & Healy (2008), to whom you also refer, show how to incorporate overprecision into an expected utility framework, and derive predictions about the preferred majority threshold of a risk-averse and overconfident (over-precise) decision maker. Here I think that enriching your literature review by acknowledging the existence of some studies on overconfidence and risk aversion would better highlight your focus on overconfidence and ambiguity. For the latter, as you also state in the paper, I am aware of no study who focuses on this specific, although very important issue. Furthermore, you are able to measure the three types of overconfidence (i.e., overestimation, overplacement and overprecision) in the same experiment, which is something that none of the above-cited papers does in decision under risk. This should also be acknowledged as one of your contributions, when introducing these papers in the literature review.
Page 5, line 191. Replace “Choi et al” with “Choi et al.”
Page 5, line 200. I guess that here “two different sources” refer to internal and external ambiguity aversion. However, when reading the previous sentence it seems that they refer to ‘willingness-to-bet’ and ‘willingness-to-invest’ paradigms. Please, rephrase the previous and/or the following sentence in order to avoid this potential misunderstanding. Furthermore, to my understanding of what follows, in the Principal Component Analysis you only consider (external) ambiguity measured within the ‘willingness-to-bet’ paradigm (since you can only use dichotomic variables in such analysis). If I am right, then you should explicitly indicate that in the Principal Component Analysis you do not consider ambiguity elicited through the ‘willingness-to-invest’ paradigm.
Page 5, lines 222-223. Replace “couples of lotteries” with “pairs of lotteries”, apart if with “couple” you mean something different than “pair”. Indeed, when reading the detailed description of Couple 1 and Couple 2 in Section 4 (page 13, lines 470-488), I did not detect anything that makes the definition of “couple of lotteries” different than the more standard “pair of lotteries” (and in fact in your experimental instructions you use the wording “Pair 1” and “Pair 2” – look at line 658 of page 17, and at line 674 of page 18, respectively). Is “couple of lotteries” the original wording used in the investment gamble of Gneezy & Potters (1997)? If yes, then keep “couple of lotteries”, and just tell the reader in a footnote that it has the same meaning of “pair of lotteries”.
Page 5, line 225. Same as comment #7.
Page 5, line 230. Here you talk about four stages of the experiment, while instead only three stages were explicitly described above. I guess that stage 4 is the one where “The experimental tokens accumulated in the first three stages of the experiment constitute the endowment subjects have the possibility of investing in two couples of lotteries involving ambiguity and confidence.” (lines 224-228) If I am right, then you should only indicate at line 224 that this is “Stage 4”.
Page 6, line 234. I find as non-standard the fact that you report the detailed description of the different tasks of the four stages of the experiment in Section 4, rather than directly describing them here, i.e. just after the short presentation of the experimental design at lines 208-233. Maybe this is a section organization feature imposed by the format of this Journal. If not, I suggest you to move Section 4 (Material and Methods) here, just before the Results section. It would then become Section 2. This is the order I have followed to read your manuscript.
Page 12, line 403. Were instructions of each stage handed to subjects and read aloud only prior to that stage? Or were they handed to subjects all together for all stages at the beginning of the experiment? This should be specified here.
Page 12, lines 413-463 (Stages 2 and 3). Sometimes it is not easy to get which of the tasks of each stage of the experiment are used to measure overestimation, overplacement and overprecision, respectively. For example, does ex-ante estimation at line 461 is a/the measure for overestimation? Does ex-post placement at line 444 is a/the measure for overplacement? Does relative precision at line 436 is a/the measure for overprecision? I think that you should clearly indicate which question/task is meant to elicit overestimation, overplacement and overprecision, at the moment where you introduce each of these questions/tasks.
Page 13, lines 464-468. At the end of Section 4 (page 14), you say nothing about the payment protocol in your experiment. I acknowledge that one can infer this information by looking at lines 464-468 (Stage 4), but I think that it would be more helpful to the reader to have all this explained more thoroughly at the end of Section 4, at page 14, after lines 493-499. There I think you should also discuss in a few lines (or in a footnote) whether and how your payment protocol is in accordance with the methodological suggestions provided by Cox, Sadiraj & Schmidt (2015), meant to minimize distortions in the elicitation of risk attitudes (in their case), that may easily extend to the case of elicitation of ambiguity attitudes (in your case). For example, you pay all earnings in the first three stages (which they say it’s a good design choice: pay-all-sequentially), but then in the last stage the computer randomly selects one of the two pairs of lotteries (which they say it’s the design choice that creates much distortion: pay-one-randomly). Here I am not saying that your payment protocol is wrong: personally, I think that every payment protocol in individual decision-making experiments with more than one decision task has pros and cons (and your experiment has four stages, with more than one task in at least one of these stages). Rather, I am just asking you to frame your payment protocol within the well-known classification of Cox, Sadiraj & Schmidt (2015) and, in the light of this classification, quickly discuss its pros and cons.
Additional References
• Attanasi, G., Corazzini, L. and F. Passarelli (2017), Voting as a lottery. Journal of Public Economics, 146, 129-137.
• Attanasi, G., Corazzini, L., Georgantzis, N., and F. Passarelli (2014). Risk aversion, overconfidence and private information as determinants of majority thresholds. Pacific Economic Review, 19, 355-386.
• Attanasi, G., Gollier, C., Montesano, A., and N. Pace (2014). Eliciting ambiguity aversion in unknown and in compound lotteries: A smooth ambiguity model experimental study. Theory and Decision, 77, 485-530.
• Cox, J. C., Sadiraj, V., and Schmidt, U. (2015). Paradoxes and mechanisms for choice under risk. Experimental Economics, 18, 215-250.
• Gysler, M., Brown Kruse, J., and Schubert, R. (2002). Ambiguity and gender differences in financial decision making: An experimental examination of competence and confidence effects. Working papers/WIF, 2002(23).
• Klibanoff, P., Marinacci, M., and Mukerji, S. (2005). A smooth model of decision making under ambiguity. Econometrica, 73, 1849-1892.
• Shyti, A. (2013). Over-confidence and entrepreneurial choice under ambiguity, Typescript.
• Yang, X., & Zhu, L. (2016). Ambiguity vs risk: An experimental study of overconfidence, gender and trading activity. Journal of Behavioral and Experimental Finance, 9, 125-131.
Please, look at the attached report.

Author Response
Please see the attachment.
Responses to Reviewer 1’s comments:
1. Page 1, line 34. Replace “knowledge,e” with “knowledge,”
Done, thank you.
2. Page 1, lines 36-42. You say that very few studies address the issues of ambiguity and confidence jointly, and you only cite Brenner et al. (2011). I am aware of three other studies that contribute to this specific field: Gysler, Brown Kruse & Schubert (2002), Shyti (2013), and Yang & Zhu (2016). I acknowledge that two of them are unpublished and that they all focus on financial or entrepreneurial decisions. Despite that, I think that you might quickly discuss in which regards these three studies relate or not to your work.
We added a discussion of these papers’ findings: please see lines 55-65.
3. Page 2, lines 52-54. The way you relate ambiguity to confidence, besides being coherent with Abdellaoui et al. (2011) source of uncertainty, is also coherent with Attanasi, Gollier, Montesano & Pace (2014) belief elicitation under ambiguity. The latter generate ambiguity in a two-stage setting à la Klibanoff, Marinacci & Mukerji (2005) by relying on different “external” sources in the first stage (i.e., binomial distribution, uniform distribution, unknown distribution) at a between-subject level. Each external source is generated by a common mechanism of uncertainty at a within-subject level. Then, given the second-stage urn composition (which is unknown), they elicit subjects’ beliefs about it. The heterogeneous distribution of beliefs they obtain can be interpreted as an “internal” source of ambiguity, i.e., in your language, different levels of “confidence” for different subjects. They find no correlation between this “confidence” and subjects’ behavior under ambiguity. I think that reporting also this study would strengthen the relation between your study and the previous experimental literature.
We now cite the paper above and discuss its relation with our investigation, please see lines 78-86.
4. Page 2, lines 64-67. I agree with you that the effects of overconfidence (i.e., overestimation, overplacement or overprecision) on ambiguity attitudes has never been previously studied in the literature (to the best of my knowledge). However, I am aware of two articles that have studied the effects of overconfidence on risk attitudes. Both these studies focus on voting choice, modeling it as a risky lottery. The first one – Attanasi, Corazzini, Georgantzis & Passarelli (2014) – studies subjects’ overconfidence and its relation with risk aversion in a theory-driven experiment where each subject can get an informative signal about the others’ voting preferences. The second one – Attanasi, Corazzini & Passarelli (2017) – explicitly refers to Moore & Healy (2008), to whom you also refer, show how to incorporate overprecision into an expected utility framework, and derive predictions about the preferred majority threshold of a risk-averse and overconfident (over-precise) decision maker. Here I think that enriching your literature review by acknowledging the existence of some studies on overconfidence and risk aversion would better highlight your focus on overconfidence and ambiguity. For the latter, as you also state in the paper, I am aware of no study who focuses on this specific, although very important issue. Furthermore, you are able to measure the three types of overconfidence (i.e., overestimation, overplacement and overprecision) in the same experiment, which is something that none of the above-cited papers does in decision under risk. This should also be acknowledged as one of your contributions, when introducing these papers in the literature review.
We now discuss the two papers above and acknowledge the specificities of our contribution with respect to the existing experimental literature, please see lines 99-105, 96-98, 35 and following.
5. Page 5, line 191. Replace “Choi et al” with “Choi et al.”
Done, thank you.
6. Page 5, line 200. I guess that here “two different sources” refer to internal and external ambiguity aversion. However, when reading the previous sentence it seems that they refer to ‘willingness-to-bet’ and ‘willingness-to-invest’ paradigms. Please, rephrase the previous and/or the following sentence in order to avoid this potential misunderstanding. Furthermore, to my understanding of what follows, in the Principal Component Analysis you only consider (external) ambiguity measured within the ‘willingness-to-bet’ paradigm (since you can only use dichotomic variables in such analysis). If I am right, then you should explicitly indicate that in the Principal Component Analysis you do not consider ambiguity elicited through the ‘willingness-to-invest’ paradigm.
We rephrased the sentence mentioned, please see line 238. We also clarified which measures we use in the Principal Component Analysis, please see lines 375-378.
7. Page 5, lines 222-223. Replace “couples of lotteries” with “pairs of lotteries”, apart if with “couple” you mean something different than “pair”. Indeed, when reading the detailed description of Couple 1 and Couple 2 in Section 4 (page 13, lines 470-488), I did not detect anything that makes the definition of “couple of lotteries” different than the more standard “pair of lotteries” (and in fact in your experimental instructions you use the wording “Pair 1” and “Pair 2” – look at line 658 of page 17, and at line 674 of page 18, respectively). Is “couple of lotteries” the original wording used in the investment gamble of Gneezy & Potters (1997)? If yes, then keep “couple of lotteries”, and just tell the reader in a footnote that it has the same meaning of “pair of lotteries”.
We replaced “couple” with “pair”, thank you.
8. Page 5, line 225. Same as comment #7.
We replaced “couple” with “pair” in this section as well.
9. Page 5, line 230. Here you talk about four stages of the experiment, while instead only three stages were explicitly described above. I guess that stage 4 is the one where “The experimental tokens accumulated in the first three stages of the experiment constitute the endowment subjects have the possibility of investing in two couples of lotteries involving ambiguity and confidence.” (lines 224-228) If I am right, then you should only indicate at line 224 that this is “Stage 4”.
We clarified that Stage 4 is represented by the set of investment choices described at the beginning of line 262.
10. Page 6, line 234. I find as non-standard the fact that you report the detailed description of the different tasks of the four stages of the experiment in Section 4, rather than directly describing them here, i.e. just after the short presentation of the experimental design at lines 208-233. Maybe this is a section organization feature imposed by the format of this Journal. If not, I suggest you to move Section 4 (Material and Methods) here, just before the Results section. It would then become Section 2. This is the order I have followed to read your manuscript.
This choice was made to respect the format of this journal.
11. Page 12, line 403. Were instructions of each stage handed to subjects and read aloud only prior to that stage? Or were they handed to subjects all together for all stages at the beginning of the experiment? This should be specified here.
Instructions were handed in written form to subjects all together for all stages at the beginning of the experiment. The instructions of each stage were also presented on subjects’ computer screen at the beginning of the stage. We now specify this in the paper, see lines 457-460.
12. Page 12, lines 413-463 (Stages 2 and 3). Sometimes it is not easy to get which of the tasks of each stage of the experiment are used to measure overestimation, overplacement and overprecision, respectively. For example, does ex-ante estimation at line 461 is a/the measure for overestimation? Does ex-post placement at line 444 is a/the measure for overplacement? Does relative precision at line 436 is a/the measure for overprecision? I think that you should clearly indicate which question/task is meant to elicit overestimation, overplacement and overprecision, at the moment where you introduce each of these questions/tasks.
We now specify the measure of overconfidence we refer to, please see lines 491, 497, 505.
13. Page 13, lines 464-468. At the end of Section 4 (page 14), you say nothing about the payment protocol in your experiment. I acknowledge that one can infer this information by looking at lines 464-468 (Stage 4), but I think that it would be more helpful to the reader to have all this explained more thoroughly at the end of Section 4, at page 14, after lines 493-499. There I think you should also discuss in a few lines (or in a footnote) whether and how your payment protocol is in accordance with the methodological suggestions provided by Cox, Sadiraj & Schmidt (2015), meant to minimize distortions in the elicitation of risk attitudes (in their case), that may easily extend to the case of elicitation of ambiguity attitudes (in your case). For example, you pay all earnings in the first three stages (which they say it’s a good design choice: pay-all-sequentially), but then in the last stage the computer randomly selects one of the two pairs of lotteries (which they say it’s the design choice that creates much distortion: pay-one-randomly). Here I am not saying that your payment protocol is wrong: personally, I think that every payment protocol in individual decision-making experiments with more than one decision task has pros and cons (and your experiment has four stages, with more than one task in at least one of these stages). Rather, I am just asking you to frame your payment protocol within the well-known classification of Cox, Sadiraj & Schmidt (2015) and, in the light of this classification, quickly discuss its pros and cons.
In Stages 1-3, subjects’ payment depends on outcome of each lottery they decide to face. Thus, each chosen option is paid before a subsequent decision is made. For what concerns Stage 4 only, subjects have to make decisions regarding ten scenarios, and only one of them is selected at random and actually implemented. According to Cox et al. (2015)’s classification, our payment protocol refers to the case where the experimenter “pays all sequentially” in Stages 1-3, and “randomly pays one decision for each subject’’ in Stage 4. Subjects’ revealed risk preferences are shown in Cox et al. (2015)’s paper to differ across mechanisms. Nonetheless, both the mechanisms we chose are demonstrated to be incentive compatible. We discuss this point at lines 559-563

Reviewer 2 Report
The authors aim to examine different/similar effects of ambiguity and confidence attitudes on risky decision behavior. To disentangle the effects the two attitudes might have on behavior, the authors design a multi-phase experiment, with multiple steps and rounds in some phases. The aim to identify three types of overconfidence (overestimation, overplacement, and overprecision). Additionally, they disentangle internal vs. external ambiguity.
In Phase 1, subjects choose between two lotteries. One lottery is based on the chance a yellow ball is drawn from a bingo cage where the distribution of red, yellow, and blue balls is ambiguous. The other lottery has a defined probability distribution.
In Phase 2 of the "Baseline" treatment, subjects first choose one of four categories for Questionnaire A (sports, entertainment, history, literature). Before taking Questionnaire A, subjects are asked to predict how many questions (out of 20) they'll answer correctly and how many other subjects in the session chose their same topic. They next moved onto to Questionnaire B. The topic for Questionnaire B is General Culture and is the same for all subjects. As with Questionnaire A, before taking Questionnaire B, subjects were asked to predict how many questions they'll answer correctly. Subjects receive token rewards for each correct multiple choice question they answer. They also receive token rewards for their predictions being within +/- 1 the actual outcomes.
In Round 1 of Phase 3, after taking the Questionnaire in Phase 1, subjects are asked to predict how many questions from Questionnaire A they accurately answered. Without feedback, subjects were then asked to choose between Lotteries Alpha and Beta. The probability a subject wins Lottery Alpha is based on his/her relative performance compared to other subjects in the session. Lottery Beta has clearly defined, independent-across-subjects probabilities of winning. Finally, subjects were asked to predict how closely their score from Questionnaire A compared to other subjects in the same session who chose the same lottery. Subjects were paid according to being within +/- 1 of the actual average.
Round 2 of Phase 3 followed in a similar manner as Round 1, except with reference to Questionnaire B.
Round 3 of Phase 3: subjects were asked to forecast the specific temperature at noon in NYC on a specific date. Subjects were then asked to choose between Lottery Epsilon (similar in flavor to Alpha?) and Lottery Zeta (similar in flavor to Beta?). Finally, subjects were asked to predict how their prediction compared to the average prediction of the other subjects in the session. Subjects were paid according to being within +/- 1 of the actual average.
Round 4 of Phase 3: subjects are asked to predict how many yellow balls are in the "ambiguous" bingo cage from Phase 1, choose between Lottery Eta (similar in flavor to Alpha?) and Lottery Iota (similar in flavor to Beta?), and finally, to predict how their guess compared to others in the session that day.
Phase 4: subjects receive information on their total earnings for the first three phases, summarized by "endowment D". They are not told details about the accuracy of any specific answers or predictions, only aggregate token earnings. Then subjects determine how to divide their D tokens in 2 Pairs of Lotteries (1 & 2). One of the two lotteries is randomly chosen for final payments. The probability of winning the prize in Lottery 1a is determined by a subject’s relative ranking within the session based on their answers to Questionnaire B. The probability of winning the prize in Lottery 2a is determined by a 50/50 draw. Lottery 1b and 2b is the same in each lottery pair, the probability that a yellow ball is drawn from the ambiguous urn.
The "No self selection" treatment was identical to the "Baseline" treatment, except that rather than choosing one of the four topics for Questionnaire A in Phase 2, the computer randomly assigned each subject a topic.
Using Principal Component Analysis, the authors find that more confident subjects also tolerate ambiguity stemming from context. Further, more-skilled subjects in the questionnaire are more ambiguity-averse (i.e., less confident) when the ambiguity is from an internal source. There are no treatment differences, self-selection does not impact behavior. Finally, subjects who rated the experiment as "easy" are less ambiguity averse and subjects who earned more money were more ambiguity averse.
That is a complex design to explain. I have a number of comments for the authors to respond to, itemized below. My biggest concern is that the novelty of the paper is not clear, especially in the introduction.
Major Comments
1. In the introduction, the authors need to better motivate and explain the novelty of their paper. If the Principal Component Analysis is a key novelty of the paper, this should be explained clearer.
2. Related to the first point, when explaining the experimental design in the introduction (Lines 195-228), the authors should explain which parts of the design are novel and which are replication/reuses of earlier studies on the topic.
3. It might be personal preference, but I don’t prefer that the design section is relegated to the end. I think the paper could benefit from restructuring with an Intro, Lit Review & Conjectures, Experimental Design to test the Conjectures, Results presented primarily around the Conjectures, and Conclusion section. As the paper reads now, the experiment was most exploratory in nature, rather than testing clear theoretical predictions. This is fine, but Conjectures at least based on other empirical results are needed.
4. The Conclusions section should be expanded to place the results in the context of the existing literature, remind the reader of the paper’s novelty, and offer suggestions for future research.
5. The treatment labels of Baseline and No Self Selection are confusing. Why not call Baseline “Self Selection”?
6. For each test, the authors should report the sample size and power. Also, for the t-tests, why not Wilcoxon signed rank tests? G*Power is a helpful online tool for calculating power for tests.
7. What is the value added for Phase 4, Investment Framing? This should be made clearer early in the paper.
8. (Table 7) With so many Phases, Rounds, etc. in the experiment, it’s easy for the reader to forget which decisions qualify for each row in the Table. For instance, which decision data is being used for Preference for low-competence-based ambiguity? Perhaps an additional column or footnote could be included to display this information.
9. (Section 4.1) Related to the point above, as each decision task is explained it should be linked back to the theoretical type of overconfidence. Also, individual decision tasks and lotteries should be linked back to previous studies or expressed as novel to this paper.
10. What information exactly is shown to subjects at the beginning of Phase 4? Based on their earnings, it seems that subjects would be able to Bayesian update their prior beliefs bore making decisions in Lottery Pairs 1 & 2. A screen shot might be helpful.
11. How do the key findings relate to findings in other studies, in particular those mentioned in the Conclusion?
Minor Comments
1. (Line 38) Brenner et al. (2011) is not in the references. Check that all other references are accurately cited and in the reference list.
2. (Lines 60-62) How do the categories relate to overconfidence ex-ante vs. ex-post decision task?
3. (Line 218) Clarify that the decisions are incentivized.
4. (Line 237) Clarify what is meant by risk, i.e., 50/50?
5. Do you have an explanation as to why subjects chose History and Literature for Questionnaire A at a much higher rate than Showbiz? Were these typical college students? Cite other papers that use a similar selection design. Are the same four categories used in other studies?
6. (Line 387) What is meant by context?
7. (Line 389) Do you mean Questionnaire A, B, or A&B?
8. (Line 404) Is 60% a good rate for this type of experiment? Cite other students that asked a similar question. I wonder if low ambiguity averse subjects could still be confused and not realize it.
9. Clearly explain how the probabilities of Epsilon and Eta are calculated. Are they calculated in a similar way as Alpha?
10. Final Questionnaire #7 (Lines 495-497 & 696-700) Explain how altruism is interpreted from the three options.
11. In the instructions the authors use Phase and in Section 4 they use Stage. They should use consistent terminology as in the instructions given to the subjects.
Author Response
Please see the attachment.
Responses to Reviewer 2’s comments:
Major Comments
1. In the introduction, the authors need to better motivate and explain the novelty of their paper. If the Principal Component Analysis is a key novelty of the paper, this should be explained clearer.
The introduction has been enlarged in order to emphasize the novelty of our contribution, see in particular lines 35-47. The Principal Component Analysis is now mentioned as a key aspect of novelty of the paper.
2. Related to the first point, when explaining the experimental design in the introduction (Lines 195-228), the authors should explain which parts of the design are novel and which are replication/reuses of earlier studies on the topic.
The experimental framework we introduce is novel; no similar design has been used before to address the issue of ambiguity and confidence jointly.
3. It might be personal preference, but I don’t prefer that the design section is relegated to the end. I think the paper could benefit from restructuring with an Intro, Lit Review & Conjectures, Experimental Design to test the Conjectures, Results presented primarily around the Conjectures, and Conclusion section. As the paper reads now, the experiment was most exploratory in nature, rather than testing clear theoretical predictions. This is fine, but Conjectures at least based on other empirical results are needed.
The design section is located at the end of the paper as a consequence of the format imposed by this journal.
4. The Conclusions section should be expanded to place the results in the context of the existing literature, remind the reader of the paper’s novelty, and offer suggestions for future research.
The Conclusions have now been rewritten in order to address your remarks, please see Section 3.
5. The treatment labels of Baseline and No Self Selection are confusing. Why not call Baseline “Self Selection”?
We renamed the Baseline as “Self Selection”, thank you for the suggestion.
6. For each test, the authors should report the sample size and power. Also, for the t-tests, why not Wilcoxon signed rank tests? G*Power is a helpful online tool for calculating power for tests.
For each test summarized in Tables 1 and 5, we now report power (last column) and sample size (in the Note below the table).
7. What is the value added for Phase 4, Investment Framing? This should be made clearer early in the paper.
The Investment Framing, presented in Phase 4 (now called Stage 4 in order to address minor comment #11), allows us to test whether subjects’ attitude towards ambiguity is affected by the specific framing (choice between pair of lotteries versus investment decisions) we adopt. This aspect is now discussed at lines 41-43.
8. (Table 7) With so many Phases, Rounds, etc. in the experiment, it’s easy for the reader to forget which decisions qualify for each row in the Table. For instance, which decision data is being used for Preference for low-competence-based ambiguity? Perhaps an additional column or footnote could be included to display this information.
The data we use for the low-competence-based ambiguity are the ones related to Questionnaire B (and data related to Questionnaire A for high-competence-base ambiguity). We add a note below Table 7 to clarify this aspect.
9. (Section 4.1) Related to the point above, as each decision task is explained it should be linked back to the theoretical type of overconfidence. Also, individual decision tasks and lotteries should be linked back to previous studies or expressed as novel to this paper.
We now refer to the specific theoretical type of overconfidence, see lines 491, 497, 505 (in brackets).
10. What information exactly is shown to subjects at the beginning of Phase 4? Based on their earnings, it seems that subjects would be able to Bayesian update their prior beliefs bore making decisions in Lottery Pairs 1 & 2. A screen shot might be helpful.
In Stage 4, subjects receive information on their earnings. Earnings depend not only on the scores in the questionnaires (from which they could infer their performance) but also in a number of other aspects related to the outcomes of the lotteries they chose and played. This obfuscates the relationship between earnings and performance in the questionnaire. Thus, we can exclude this process of Bayesian update of beliefs could drive our results.
11. How do the key findings relate to findings in other studies, in particular those mentioned in the Conclusions?
Please, see the new version of the Conclusions.
Minor Comments
1. (Line 38) Brenner et al. (2011) is not in the references. Check that all other references are accurately cited and in the reference list.
Brenner et al. (2011) is now in the references, thank you.
2. (Lines 60-62) How do the categories relate to overconfidence ex-ante vs. ex-post decision task?
These categories are orthogonal to the distinction between ex-ante and ex-post decision tasks. Moore and Healy (2008)’s classification deals with the way how overconfidence is measured. Ex-ante versus ex-post decision tasks differ on measuring confidence before or after completing the task and can be be associated either to overestimation, overplacement or overprecision as in Moore and Healy (2008)’s classification.
3. (Line 218) Clarify that the decisions are incentivized.
Done, thank you, see line 257.
4. (Line 237) Clarify what is meant by risk, i.e., 50/50?
Yes, we meant 50-50. It has been clarified in the present version of the paper (line 250), thank you.
5. Do you have an explanation as to why subjects chose History and Literature for Questionnaire A at a much higher rate than Showbiz? Were these typical college students? Cite other papers that use a similar selection design. Are the same four categories used in other studies?
Our participants were typical college students. We have no explanation for observing a lower preference for the “Showbiz” category and there is no previous study presenting the same four categories to understand whether this is a “typical” behavior or not. Our speculation is that perhaps this topic could appear less standardized with respect to History and Literature and thus be perceived as more challenging.
6. (Line 387) What is meant by context?
Context refers to something “external” to the subject.
7. (Line 389) Do you mean Questionnaire A, B, or A&B?
There is no Questionnaire mentioned at line 389 of the previous version of the paper, so we were not able to address this point, we are sorry.
8. (Line 404) Is 60% a good rate for this type of experiment? Cite other students that asked a similar question. I wonder if low ambiguity averse subjects could still be confused and not realize it.
We are not aware of other studies asking to evaluate the easiness of the experiment by participants. We decided to add this question because we wanted to evaluate how our subjects perceived the experiment and how this evaluation could relate to their behavior.
9. Clearly explain how the probabilities of Epsilon and Eta are calculated. Are they calculated in a similar way as Alpha?
Yes, they are calculated in a similar way as Alpha, we now specify this in lines 688 and next, 697 and next.
10. Final Questionnaire #7 (Lines 495-497 & 696-700) Explain how altruism is interpreted from the three options.
In the Final Questionnaire (question #7), a subject is classified as behaving altruistically if she selects the answer “performing in line with the others”. We now specify this in line 763.
11. In the instructions the authors use Phase and in Section 4 they use Stage. They should use consistent terminology as in the instructions given to the subjects.
We now use “Stage” both in the paper and in the instructions, thank you.
